# Enhance Eye Disease Detection using Learnable Probabilistic Discrete Latents in Machine Learning Architectures

## Abstract

Ocular diseases, including diabetic retinopathy and glaucoma, present a significant public health challenge due to their high prevalence and potential for causing vision impairment. Early and accurate diagnosis is crucial for effective treatment and management. In recent years, deep learning models have emerged as powerful tools for analysing medical images, such as retina imaging. However, challenges persist in model relibability and uncertainty estimation, which are critical for clinical decision-making. This study leverages the probabilistic framework of Generative Flow Networks (GFlowNets) to learn the posterior distribution over latent discrete dropout masks for the classification and analysis of ocular diseases using fundus images. We develop a robust and generalizable method that utilizes GFlowOut integrated with ResNet18 and ViT models as the backbone in identifying various ocular conditions. This study employs a unique set of dropout masks - none, random, bottomup, and topdown - to enhance model performance in analyzing these fundus images. Our results demonstrate that our learnable probablistic latents significantly improves accuracy, outperforming the traditional dropout approach. We utilize a gradient map calculation method, Grad-CAM, to assess model explainability, observing that the model accurately focuses on critical image regions for predictions. The integration of GFlowOut in neural networks presents a promising advancement in the automated diagnosis of ocular diseases, with implications for improving clinical workflows and patient outcomes. The source code for all these experiments can be found at https://github.com/anirudhprabhakaran3/gflowout_on_eye_images.

## 1 Introduction

The world faces considerable challenges in terms of eye care. Ocular diseases represent serious public health challenges due to their widespread prevalence (Zhou et al., 2023) and potential to cause significant vision impairment (Yang et al., 2021). Studies project that in 2040, there will be an estimated 288 million global cases of age-related macular degeneration (Wong et al., 2014). According to the World Health Organization, in its report of titled "World Report on Vision", more than 2.2 billion people suffer from vision impairment or blindness. Importantly, it is estimated that over 1 billion of these cases could potentially have been avoided with proper prevention or effective treatment. It indicates that primary causes of blindness include Glaucoma, Age-Related Macular Degeneration, and Diabetic Retinopathy. Diagnosing these conditions typically involves an ophthalmologist evaluating a patient's symptoms, analyzing various eye and retina images, and conducting a manual examination. This process is thorough but can be time-consuming (WTO, 2019). Other researchers have highlighted that the prevalence of Age-Related Macular Degenerations (AMDs) is notably higher in Africa and the Eastern Mediterranean regions compared to other areas of the world (Xu et al., 2020). The lack of and unequal distribution of medical resources means that preventable and treatable cases of blindness and low vision predominantly affect people in less developed countries and regions. Vision impairment stems from various factors, notably the retina, which is a key element in disorders like glaucoma, diabetic retinopathy, and age-related macular degeneration. Properly addressing eye health requires not only accurate diagnosis but also effective prevention and treatment strategies for these conditions (Yang et al., 2021).

Ophthalmology heavily depends on imaging for diagnosis, as the majority of eye conditions are identified through image analysis. However, traditional screening involves handling large volumes of data, is highly subjective, and requires complex data analysis. This presents a significant burden for both patients and ophthalmologists, complicating long-term follow-ups (Besenczi et al., 2016). Early diagnosis and effective management are crucial in preventing these diseases from progressing to more severe stages. Traditional diagnostic methods, which typically involve manual examination of retinal images, are often time-consuming and subject to variability among practitioners. The incorporation of artificial intelligence (AI), particularly machine learning and deep learning, into this field has significantly boosted the efficiency of clinical eye specialists. AI technology processes and analyzes ophthalmic images, thereby streamlining diagnostic procedures (Padhy et al., 2019; Yu et al., 2018; Rajpurkar et al., 2022; Zhou et al., 2023). Since 2016, Google has applied Deep Learning to analyze retinal images for the detection of diabetic retinopathy (Gulshan et al., 2016). Currently, there has been considerable research on artificial intelligence-assisted diagnosis in diseases such as glaucoma, diabetic retinopathy, retinopathy of prematurity, and age-related macular degeneration (AMD) (Ting et al., 2018). The utilized of Deep Learning has shown immense promise in automating the analysis of medical images, providing more consistent and scalable solutions for disease diagnosis. However, we found that most of the models focus primarily on diagnosing a single ophthalmic disease (Li et al., 2021). There are multiple works showing that deep learning algorithms are promising in the diagnosing diabetic retinopathy through retinal fundus image grading (Oh et al., 2021; Wang et al., 2022; Son et al., 2020). However, the high performance of these methods often comes with a significant increase in time complexity. Additionally, issues around the reliability of these models and their capacity to estimate uncertainty continue to present challenges in clinical decision-making (Li et al., 2022).

Along with these issues, a significant limitation of current deep neural networks is their tendency to exhibit *overconfidence in predictions* and lack a mechanism for capturing uncertainty, particularly when there is a shift in the data distribution between training and testing datasets (Folgoc et al., 2021). This issue is especially prominent in medical imaging, where variability in data can impact diagnostic accuracy. While methods such as standard dropout exist to address this, they often fail to capture the multi-modality of posterior distributions over dropout masks. To mitigate these challenges, GFlowOut (Liu et al., 2023) has been recently proposed, leveraging Generative Flow Networks (GFlowNets) (Bengio et al., 2023) to model the posterior distribution over dropout masks. However, its potential in real-world medical applications remains underexplored. Our key research question is: How can probabilistic dropout masks be leveraged to enhance the performance of deep learning models in ocular disease detection? We explore Generative Flow Networks as a probabilistic framework to address these limitations. Specifically, we model the posterior distribution over discrete dropout masks to improve classification and analysis of ocular diseases using retinal fundus images. We explore Generative Flow Networks as a probabilistic framework to address these limitations. Specifically, we model the posterior distribution over discrete dropout masks to improve classification and analysis of ocular diseases using retinal fundus images.

Figure 1 illustrates the integration of GFlowOut within the Vision Transformer architecture for eye disease detection. The input fundus image is divided into patches, which are processed through the transformer layers to extract feature representations. In place of traditional dropout layers, GFlowOut layers are applied to model the posterior distribution over discrete dropout masks, enhancing the model's uncertainty estimation and robustness. The output from the GFlowOut layers is passed through multi-layer perceptron (MLP) heads, which map the processed features to the predicted disease class. This architecture demonstrates the use of learnable probabilistic latents to improve model reliability and diagnostic performance, as detailed in Section 3.1.

Our key contribution is demonstrating the utility of GFlowOut in improving model uncertainty estimation and diagnostic accuracy across a diverse set of ocular conditions, providing a robust solution to variability in medical imaging datasets. The integration of GFlowOut into neural networks not only improves model reliability and accuracy but also contributes to the development of more reliable diagnostic tools that can assist in clinical workflows, potentially enhancing patient outcomes.

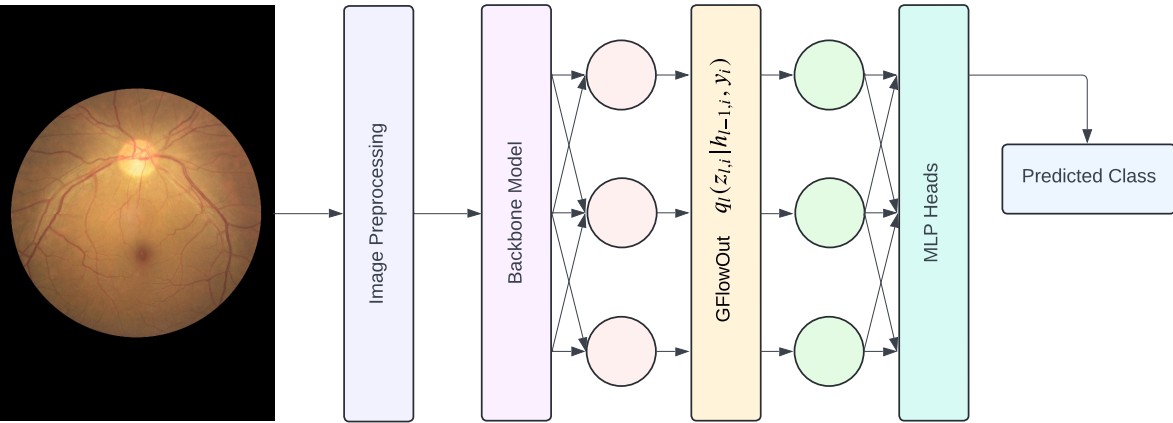

Figure 1: In the vision transformer architecture, we apply GFlowOut, a learnable dropout technique, in the transformer encoder. This allows us to learn posterior distribution over dropout masks tailored to our dataset, improving performance of the model.

## 2 Related Work

### 2.1 Generative Flow Networks

Generative Flow Networks (GFlowNets) have recently emerged as a compelling framework for generating complex, high-dimensional objects by modeling the flow of probability through sequences of states. GFlowNets address the challenge of sampling objects in proportion to a predefined reward function by adopting a control problem formulation, where objects are constructed sequentially via probabilistic steps. This methodology enables GFlowNets to efficiently explore and sample from multimodal distributions, making them particularly well-suited for applications requiring diverse and high-quality solutions, such as drug discovery and protein design (Bengio et al., 2023).

The versatility of GFlowNets has been demonstrated across various domains, including drug discovery (Bengio et al., 2021a), biological sequence design (Jain et al., 2022), robust combinatorial optimization (Zhang et al., 2022), causal discovery (Deleu et al., 2022), and neural network structure learning (Pan et al., 2023a). Foundational work has highlighted the ability of GFlowNets to generalize effectively to complex distributions and reduce gradient variance relative to traditional policy gradient methods, thereby establishing a robust framework for probabilistic modeling (Malkin et al., 2023b; Bengio et al., 2021b).

Subsequent advancements have further extended the capabilities of GFlowNets. For instance, (Pan et al., 2023b) introduced Stochastic GFlowNets to address the challenges posed by stochastic environments, incorporating intrinsic exploration rewards to enhance training. Additionally, (Deleu et al., 2022; Nishikawa-Toomey et al., 2022) applied GFlowNets to the generative modeling of discrete and composite objects, with a particular focus on Bayesian structure learning of complex causal graphs. The framework has also been leveraged in approximate maximum-likelihood training of energy-based models, as demonstrated by (Zhang et al., 2022), without the need for a predefined target reward. Moreover, GFlowNets have been applied to tackle NP-hard combinatorial optimization problems, providing a promising approach to these computationally intensive tasks (Zhang et al., 2022). In the realm of biological sequence design, (Jain et al., 2022) employed GFlowNets within an active learning loop to optimize sequence generation. Furthermore, (Zimmermann et al., 2023) offered a variational perspective on GFlowNets by formulating variational objectives through the use of KL divergences. Collectively, these studies underscore the adaptability and potential of GFlowNets in addressing a wide array of generative modeling challenges across diverse fields.

## 2.2 GFlowOut - Dropout with Generative Flow Networks

(Liu et al., 2023) introduced GFlowOut, a novel solution to the challenges inherent in traditional dropout techniques used within neural networks. These challenges include the multi-modality of the posterior distribution over dropout masks and the difficulty in fully utilizing sample-dependent information and the correlation among dropout masks. GFlowOut leverages the principles of Generative Flow Networks (GFlowNets) to enhance dropout regularization by learning the posterior distribution over dropout masks. Traditional dropout methods often struggle to accurately capture the posterior due to the multimodal and discrete nature of dropout masks (Liu et al., 2023; Jain et al., 2022).

GFlowOut addresses these limitations by employing GFlowNets to generate layer-wise dropout masks that are conditioned on previous layer activations and labels, thus improving the estimation of uncertainty and robustness to distributional shifts. Empirical evaluations have demonstrated that GFlowOut significantly outperforms standard methods, such as Random Dropout and Contextual Dropout, across a variety of tasks, including image classification under deformations, visual question answering, and real-world clinical predictions (Liu et al., 2023). By utilizing the Trajectory Balance objective during training, GFlowOut ensures that the generated masks are proportionate to the reward function, providing a robust framework for improving posterior estimation and effectively leveraging sample-dependent information in neural networks. This results in enhanced generalization and superior performance in downstream tasks (Liu et al., 2023; Malkin et al., 2023a).

# 3 Method

## 3.1 Model Structure

In our approach, we integrate learnable probabilistic discrete latent variables into established vision models by implementing GFlowOut within the architectures of ResNet18 and Vision Transformer, which serve as the backbone models. To achieve this, we modify specific layers of these models to incorporate GFlowOut functionality.

For the ResNet18 model, the standard dropout layers present after every residual block were removed by setting the dropout probability to 0. In their place, we introduced GFlowOut layers to manage the dropout process. This modification was consistently applied across all 12 residual blocks, though the implementation is flexible and can be customized to target specific blocks while omitting others as needed.

In the Vision Transformer architecture, we implemented dropout after every Attention-MLP block. Similar to our approach with ResNet18, the dropout probability for the standard dropout layers was set to 0, and GFlowOut layers were inserted to manage the dropout.

Both backbone models, ResNet18 and Vision Transformer, were pre-trained on the ImageNet dataset. Following pre-training, the final dense layers of these models were fine-tuned on the specific dataset utilized in this study. The GFlowOut layers are implemented as multi-layer perceptron (MLP) layers, which compute dropout probability distributions based on the context provided by previous layers and the input to the current layer, contingent on the masks used.

To further clarify the step-by-step implementation of this approach, Algorithm 1 outlines the detailed pipeline for eye disease detection. The process begins with standard image preprocessing, including normalization and cropping to remove irrelevant regions, followed by partitioning images into patches for efficient processing. These patches are then input into the Vision Transformer's feature extractor, leveraging self-attention to learn rich feature representations. The GFlowOut layers are applied after each Attention-MLP block to model the posterior distribution of discrete dropout masks, improving robustness against overfitting and data distribution shifts. Finally, these regularized features are passed through the classification head to predict disease probabilities for each input image. This algorithm operationalizes the proposed framework by translating the described architecture into actionable steps, showcasing the practical integration of probabilistic regularization techniques like GFlowOut into state-of-the-art vision models.

The GFlowOut layers are implemented as multiple MLPs. $q(z_i|x_i, y_i; \phi)$ is implemented as a set of multiple MLPs, once for each layer in the model needing dropout. At layer $l$, the dropout probabilities of all units in layer $l$ are estimated, and repeated for each layer in the model. For more implementation details, you can check the original GFlowOut paper (Liu et al., 2023).

---

**Algorithm 1:** Algorithm for Eye Disease Detection Using Vision Transformer and GFlowOut

**Input:** Retinal images dataset `Images`
**Output:** Disease probability `P` for each input image
**Data:** Normalized retinal images, patches of images, feature vectors, attention weights

1 **Function** process_images(*Images*):
    // Step 1: Preprocessing the retinal images
2     **foreach** *Image in Images* **do**
3         Image ← Normalize(Image) // Standardize pixel intensity across images
4         Image ← Crop(Image) // Remove irrelevant portions of the image
    // Step 2: Split the images into patches
5     **foreach** *Image in Images* **do**
6         Patches ← SplitIntoPatches(Image) // Segment image into smaller patches
    // Step 3: Pass patches through Vision Transformer feature extractor
7     **foreach** *Patch in Patches* **do**
8         Features ← ViTFeatureExtractor(Patch) // Extract feature vectors using the Vision Transformer
    // Step 4: Pass features through attention heads
9     **foreach** *Feature in Features* **do**
10         AttentionOutput ← ApplyAttentionHeads(Feature) // Use attention to highlight relevant data
    // Step 5: Apply GFlowOut for enhanced regularization
11     **foreach** *AttentionOutput in AttentionOutputs* **do**
12         RegularizedFeature ← ApplyGFlowOut(AttentionOutput) // Regularize features using learnable dropout
    // Step 6: Compute reward and trajectory balance
13     **foreach** *RegularizedFeature in RegularizedFeatures* **do**
14         Reward ← $R$(RegularizedFeature) // Compute reward based on performance metrics
15         TrajectoryBalance ← $\prod_{t=0}^{T-1} P(a_t \mid s_t) = \frac{R(s_T)}{Z}$ // Ensure trajectory balance based on reward
    // Step 7: Pass data through the classification head
16     **foreach** *RegularizedFeature in RegularizedFeatures* **do**
17         P ← ClassificationHead(RegularizedFeature) // Compute probability of disease
    // Step 8: Output probabilities
18     **foreach** *Image in Images* **do**
19         Print(P) // Display the disease probability for the input image

---

### 3.2 GFlowOut Masks

Generative Flow Networks (GFlowNets) provide a probabilistic framework to sample objects (e.g., dropout masks) in proportion to a reward function. GFlowOut integrates GFlowNets into neural networks to dynamically learn dropout masks, improving model robustness and performance in tasks such as image classification. GFlowNets generate objects by modeling the flow of probabilities through sequential states. Trajectory Balance Objective aligns mask generation with task-specific rewards.GFlowOut's dynamic mask generation adapts to data variability, improving regularization and robustness, particularly for tasks like image classification.

**Flow Consistency Equation :** The flow consistency equation ensures the flow of probabilities into and out of a state $s$ is balanced:

$$F(s) = \sum_{a \in \text{Actions}(s)} F(s, a),$$

where:

- $F(s)$ is the total flow into state $s$.

- $F(s, a)$ is the flow through state $s$ via action $a$.

**Trajectory Balance Objective :**   The trajectory balance objective ensures that the probability of generating a trajectory $\tau$ is proportional to the reward of its terminal state. For a trajectory $\tau = (s_0, a_0, s_1, \ldots, s_T)$:

$$\prod_{t=0}^{T-1} P(a_t \mid s_t) = \frac{R(s_T)}{Z},$$

where:

- $P(a_t \mid s_t)$ is the policy probability of taking action $a_t$ at state $s_t$.

- $R(s_T)$ is the reward of the terminal state $s_T$.

- $Z$ is the normalization constant (partition function).

**Dropout Mask Probability $P(\textbf{Mask})$ :**   GFlowOut learns the posterior distribution over dropout masks. The probability of a specific mask $M$ is proportional to its reward:

$$P(M) \propto R(M).$$

**Posterior Distribution :**   The posterior distribution is computed using a softmax function:

$$P(\text{dropout} \mid X, H) = \text{softmax}(W \cdot [X, H]),$$

where:

- $X$ is the input data.

- $H$ is the context from preceding layers.

- $W$ are learnable weights.

In this study, we employ four types of masks: `none`, `random`, `bottomup`, and `topdown`. The `none` mask indicates the absence of any applied mask. The `random` mask functions similarly to traditional dropout layers, applying a randomly generated mask, thereby mimicking the behavior of standard random dropout.

The `bottomup` mask generates dropout masks based on both the input data and the contextual information from previous layers, allowing for a more data-driven computation of the dropout probability distribution. In contrast, the `topdown` mask creates dropout masks solely based on the contextual information from preceding layers, without incorporating any direct input data.

We hypothesize that the `bottomup` masks will outperform the others, as they leverage additional data input to inform the computation of the dropout probability distribution, potentially leading to more effective regularization and improved model performance.

### 3.3   Eye Disease Dataset

In this study, we use three datasets - the Ocular Disease Intelligent Recognition (ODIR) dataset (Maranhão, 2020), the Retinal Fundus Multi-Disease (RFMiD) Image dataset (Panchal et al., 2023) and the Joint Shantou International Eye Center (JSIEC) (Cen et al., 2021) dataset. The ODIR and RFMiD datasets are used for training of the model and evaluation, and the JSIEC dataset is used for out-of-distribution (OOD) and uncertainty estimation experiments.

The Ocular Disease Intelligent Recognition (ODIR) dataset is a comprehensive ophthalmic database consisting of records from 5,000 patients, including age information, color fundus photographs of both eyes, and

diagnostic keywords provided by medical professionals. The uniqueness of this dataset is in providing paired data for both left and right eyes along with descriptions. However, for our experiments and model architecture, we consider the two fundus images separate, and extract the diagnosis per image from the description. ODIR-5K provides 8 categories of labels: Normal (N), Diabetes (D), Glaucoma (G), Cataract (C), Age-related Macular Degeneration (AMD), Hypertension (H), Myopia (M), and Other diseases/abnormalities (O).

The Retinal Fundus Multi-Disease (RFMiD) Image Dataset is a multi-label classification dataset for fundus images. We use the 2.0 version of the dataset for our study. The dataset contains a total of 3200 cases, with annotations for 45 different ocular diseases. Given the large number of annotation categories in the dataset, some have very few samples, making this dataset also relevant for research into multi-label long-tail problems and issues with limited samples. The Joint Shantou International Eye Center (JSIEC) dataset consists of 1000 fundus images over 39 categories. We use this subset of the bigger dataset, since we are only using the data for OOD evaluation. Table 1 presents this metadata about all the datasets used in brief.

Table 1: Metadata about the datasets used.

| Dataset Name | Data Files | Classes |
|---|---:|---:|
| ODIR | 10000 | 8 |
| RFMiD | 3200 | 45 |
| JSIEC | 1000 | 39 |

Prior to being fed into the model, the images underwent several pre-processing steps. Initially, the images were center cropped to a size of $224 \times 224 \times 3$ pixels. These images were then normalized using means of $\mu = [0.485, 0.456, 0.406]$ and standard deviations of $\sigma = [0.229, 0.224, 0.225]$. These pre-processing steps are consistent with the standard procedures for preparing inputs to ResNet and Vision Transformer models.

An important aspect that we wish to study is how well our model would perform with actual clinical data. To emulate the noises in medical imaging equipment, we add Gaussian, Salt-and-Pepper and Speckle noise. Gaussian noise can arise due to electronic noise in the imaging sensors or equipment and presents random variations in pixel intensity as fine grain-like interference across the image. Salt-and-Pepper noise causes random white and black pixels, occurring due to transmission errors, sensor defects or corruption due to image storage or transfer. Speckle noise also occurs as grainy, salt-and-pepper-like interference but has a more localised and granular pattern. This usually occurs in random interference or variations during the imaging process. In experiments using noisy data, we manually add these noises to the images after the standard pre-processing steps are completed.

### 3.4 Out of Distribution Evaluation and Entropy Calculations

Entropy measures the uncertainty or unpredictability of a probability distribution. For a single sample, the entropy $H$ is given by:

$$H = -\sum_i p_i log(p_i)$$

where $p_i$ is the predicted probability for class $i$.

Higher entropy indicates higher uncertainty, while lower entropy indicates greater confidence in the prediction. We calculate entropy on both the training dataset and the evaluation datasets. This is done to measure how confident the model is in its predictions for data. Calculating entropy on the OOD datasets allows us to evaluate robustness. A well-trained model should show higher entropy for OOD samples, including uncertainty because these samples deviate from the training distribution. It might be overconfident if a model assigns low entropy (high confidence) to OOD samples.

To get a more quantitative idea about the performance of the model, we also compute the Expected Calibration Error (ECE). Model calibration aims to align the predictions of a model with the true probabilities

and thereby making sure that the predictions of a model are reliable and accurate. Although ECE is not specifically designed for OOD detection, better calibration on in-distribution (ID) data indirectly enhances OOD detection by ensuring that confidence scores or uncertainty measures, such as entropy, are meaningful. We compute ECE on the ID dataset to assess the quality of the model's uncertainty estimation and demonstrate that our proposed method, GFlowOut, achieves significantly lower ECE compared to baseline models.

We define the ECE as

$$ECE = \sum_{m=1}^{M} \frac{|B_m|}{n} |acc(B_m) - conf(B_m)|$$

where $M$ is the number of bins, $m$ is the bin number, $|B_m|$ is the size of the bin, $conf(B_m)$ is the average estimated probabilities in bin $m$, defined as

$$conf(B_m) = \frac{1}{|B_m|} \sum_{i \in B_m} \hat{p}_i$$

and $acc(B_m)$ is the accuracy per bin $m$, defined as

$$acc(B_m) = \frac{1}{|B_m|} \sum_{i \in B_m} (\hat{y}_i = y_i)$$

A lower ECE score indicates that the model is more calibrated towards the actual probabilities.

## 4 Experiments and Results

### 4.1 Eye Disease Detection Experiment

The models were trained using NVIDIA Tesla P100 GPUs for 100 epochs. The dataset was divided into training and testing subsets with a split ratio of `0.2`, ensuring a robust evaluation framework. During the training process, both models were subjected to all four different map types, with the results tabulated for comparative analysis. Our findings indicate that the Vision Transformer generally outperforms the ResNet model. However, when focusing on the same backbone model, the `bottomup` mask emerges as the superior performer, delivering the highest accuracy among the tested configurations. Conversely, the model with no mask applied exhibited the lowest accuracy levels, underscoring the critical role of appropriate masking strategies.

We also performed experiments with noise added to the images, which revealed insightful results. Models equipped with GFlowOut showed enhanced performance compared to their standard counterparts, even under noisy conditions. We only report the results of the Vision Transformer model with `bottomup` mask, since this model performs the best on the standard datasets. Remarkably, the accuracy of these models with GFlowOut remained comparable to scenarios involving non-noisy data, underscoring the robustness of the model against different types of noise. This robustness is a significant finding, highlighting the model's potential for practical applications where data imperfections are common.

These results are in line with our expectations. The Vision Transformer, being both a larger and transformer-based model as compared to the ResNet-18 model, is expected to learn more features from the datasets and perform better at the task out of the models in consideration. Similarly, for a fixed backbone model, we expect the observed pattern in the various masks. The `none` mask performs the worst, since it is behaving as though there is no dropout. The `random` mask performs like a regular dropout layer, which is slightly better than having no dropout in these large models. `topdown` and `bottomup` perform better and the best respectively, since they take into consideration the previous layer's context, and in the case of `bottomup` mask, the input data as well, to compute the probability distribution that is to be used for dropout.

Table 2: Experimental results of disease diagnosis on the ODIR dataset. The below metrics mentioned are weighed averages. We note that the `bottomup` mask based on GFlowOut outperforms the other methods. We also note that the GFlowOut methods perform better than the baseline, and the `random` mask performs nearly the same. This is expected because this mask acts as a normal dropout.

|  |  | **Precision** | **Recall** | **F1–Score** | **Accuracy** | **AUROC** |
|---|---|---|---|---|---|---|
| | `benchmark` | 0.73 | 0.71 | 0.72 | 61.08 | 0.842 |
| | `none` | 0.73 | 0.71 | 0.72 | 61.08 | 0.798 |
| ResNet18 | `random` | 0.75 | 0.76 | 0.75 | 65.66 | 0.832 |
| | `bottomup` | **0.80** | **0.78** | **0.80** | **68.82** | **0.864** |
| | `topdown` | 0.77 | 0.69 | 0.73 | 66.67 | 0.844 |
| | `benchmark` | 0.72 | 0.58 | 0.64 | 72.04 | 0.883 |
| | `none` | 0.73 | 0.74 | 0.74 | 67.04 | 0.811 |
| Vision Transformer | `random` | 0.79 | 0.79 | 0.79 | 72.52 | 0.846 |
| | `bottomup` | **0.83** | **0.81** | **0.82** | **81.16** | **0.923** |
| | `topdown` | 0.72 | 0.71 | 0.71 | 78.19 | 0.889 |

Table 3: Experimental results of disease diagnosis on the RFMiD dataset. The below metrics mentioned are weighed averages. We note that the `bottomup` mask based on GFlowOut outperforms the other methods. We also note that the GFlowOut methods perform better than the baseline.

|  |  | **Precision** | **Recall** | **F1–Score** | **Accuracy** | **AUROC** |
|---|---|---|---|---|---|---|
| | `benchmark` | 0.94 | 0.91 | 0.93 | 89.1 | 0.947 |
| | `none` | 0.90 | 0.88 | 0.89 | 87.8 | 0.932 |
| ResNet | `random` | 0.93 | 0.91 | 0.92 | 89.4 | 0.951 |
| | `bottomup` | **0.95** | **0.95** | **0.95** | **90.3** | **0.961** |
| | `topdown` | 0.93 | 0.95 | 0.94 | 89.9 | 0.955 |
| | `benchmark` | 0.95 | 0.93 | 0.94 | 91.1 | 0.959 |
| | `none` | 0.91 | 0.88 | 0.89 | 89.8 | 0.939 |
| Vision Transformer | `random` | 0.95 | 0.94 | 0.94 | 92.0 | 0.962 |
| | `bottomup` | **0.96** | **0.95** | **0.96** | **93.1** | **0.969** |
| | `topdown` | 0.96 | 0.94 | 0.95 | 92.6 | 0.964 |

Table 4: Robustness to noise experiments. We note that the model is quite robust to noise, which was added manually to both datasets. This is an important metric since in clinical scenarios, it is quite possible that the images acquired have some sort of noise in them.

| **Dataset** | **Noise** | **Precision** | **Recall** | **F1–Score** | **Accuracy** |
|---|---|---|---|---|---|
| | Gaussian | 0.81 | 0.81 | 0.81 | 80.94 |
| ODIR | Salt | 0.76 | 0.72 | 0.74 | 75.24 |
| | Speckle | 0.72 | 0.70 | 0.71 | 69.66 |
| | Gaussian | 0.92 | 0.91 | 0.91 | 91.3 |
| RFMiD | Salt | 0.90 | 0.87 | 0.88 | 90.8 |
| | Speckle | 0.92 | 0.89 | 0.90 | 89.9 |

## 4.2 Out of Distribution Evaluation and Entropy Calculations

To thoroughly evaluate the performance of our model, we tested it on out-of-distribution (OOD) datasets and calculated the entropy of the forward pass results. Specifically, we utilized the JSIEC dataset (JSIEC) as our OOD dataset for evaluation. The JSIEC dataset, recognized for its comprehensive and diverse collection of eye images, presents significant challenges, making it an ideal benchmark for assessing the robustness and generalization capabilities of the model.

In our evaluation process, we performed multiple forward passes on both the training and evaluation datasets. By calculating the entropy of the outputs from these forward passes, we quantified the uncertainty in the model's predictions. Typically, higher entropy values indicate greater uncertainty, while lower entropy values suggest more confident predictions. By analyzing these entropy values, we identified patterns and differences in the model's performance on in-distribution versus out-of-distribution data. This analysis also enabled us to pinpoint specific images within the datasets associated with high or low entropy. Images with high entropy often highlight areas where the model struggles to make confident predictions, revealing potential weaknesses. Conversely, images with low entropy indicate areas where the model excels, making accurate and confident predictions.

Specifically, we conducted five forward passes using the ViT-GFN model on both the training and evaluation datasets. For each pass, we computed the minimum, maximum, and average entropy values. These results are systematically presented in Table 5. By examining high and low entropy images, we gained a deeper understanding of the types of data our model handles effectively and the types that pose challenges. This information is crucial for guiding future improvements and fine-tuning the model to enhance its overall performance.

To further explore the explainability of our model, we visualized the attention maps of the Vision Transformer model. Using the PyTorch GradCAM implementation (Gildenblat & contributors, 2021), we generated attention maps and overlaid these maps on the original sample images. This visualization highlights the regions of the image deemed important by the model, thereby enhancing our understanding of the model's decision-making process.

Table 5: Entropy calculations. We use the `random` mask as our baseline, as that is similar to normal dropout, and compare it with the `bottomup` mask. The models trained on these datasets were evaluated against the JSIEC dataset. We note that our model has higher uncertainty on OOD dataset than the baseline model.

| Dataset | Mask | Value |
|---------|------|-------|
| ODIR | `random` | $0.37 \pm 0.01$ |
| | `bottomup` | $0.50 \pm 0.12$ |
| RFMiD | `random` | $0.36 \pm 0.05$ |
| | `bottomup` | $0.41 \pm 0.09$ |

The entropy data provides significant insights into the model's performance. By analyzing the entropy values, we can identify which images our model handles well and which ones it struggles with. Images that exhibit the lowest entropy values, as shown in Figure 4, typically perform better. These images are often clear and well-centered, facilitating more accurate model predictions. Conversely, images with the highest entropy values, depicted in Figure 4, tend to perform worse. These problematic images are frequently either too bright or too dark, complicating the model's ability to make accurate predictions. Additionally, unclear or blurry images significantly degrade the model's performance, leading to lower accuracy rates.

In Table 6, we present the Expected Calibration Error (ECE) scores for both In-Distribution (ID) and Out-of-Distribution (OOD) datasets, evaluated using the baseline and GFlowOut models. The ECE scores are computed with $M = 10$ bins. A lower ECE score indicates better model calibration, reflecting a closer alignment between predicted probabilities and actual outcomes.

Our results demonstrate that the GFlowOut model consistently achieves lower ECE scores compared to the baseline model. This finding indicates that GFlowOut provides better-calibrated predictions on both ID and OOD datasets. The improvement in calibration is particularly notable for the OOD dataset, where the

Table 6: ECE Calculations. We note that the GFlowOut model performs better on both ID and OOD datasets.

| Dataset | Mask | ECE Score |
|---------|------|-----------|
| **ID** | random | 0.10468232 |
| | bottomup | 0.06402796 |
| **OOD** | random | 0.23651563 |
| | bottomup | 0.14311012 |

GFlowOut model outperforms the baseline model, emphasizing its robustness in handling data beyond the training distribution.

Furthermore, we observe a correlation between model uncertainty, as measured by entropy, and ECE scores. Specifically, scenarios with higher uncertainty tend to exhibit higher ECE scores. This relationship suggests that calibration performance is influenced by the model's confidence, with less confident predictions typically being less calibrated.

Finally, we computed the attention maps and superimposed them on the original sample images (Figure 5). We observed that the fundus images of diabetes patients highlighted specific vessels and areas deemed more important by the model. In contrast, the fundus images of normal patients showed more dispersed attention maps, indicating that no specific area of the image contributed predominantly to the classification output.

## 5 Conclusion

In this study, we present a novel methodology for advancing eye disease detection by integrating learnable probabilistic discrete latents via GFlowOut within ResNet18 and Vision Transformer architectures. Our approach has demonstrated substantial improvements in both accuracy and robustness, particularly under challenging conditions such as noisy data and out-of-distribution scenarios. Empirical evidence reveals that the use of bottom-up and top-down dropout masks, specifically tailored to the dataset, significantly enhances model performance, surpassing the effectiveness of conventional dropout methods. Additionally, the entropy analysis provided critical insights into the model's predictive confidence, highlighting areas for further optimization.

By enhancing the model's capacity to generalize and manage uncertainty, our approach marks a pivotal advancement in the development of reliable AI-driven diagnostic tools for clinical applications. Future research should investigate the broader applicability of this method across other medical imaging domains and focus on refining the model to improve its interpretability and clinical relevance.

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

# A Datasets

## A.1 ODIR

The ODIR-5K dataset, released by a Chinese team, is a multi-label classification dataset of fundus images. It was made available during the "Intelligent Eye" competition hosted by Peking University in 2019 and contains paired fundus images of the left and right eyes from 5000 patients, with labeled data for 3500 cases released for training.

Unlike other fundus datasets (like CHASE and DRIVE), ODIR-5K's key distinction lies in providing paired data for both left and right eyes along with relevant descriptions. This setup allows for the data to be used both as 7000 individual cases or as 3500 paired cases for exploring consistencies or other aspects. ODIR-5K provides 8 categories of labels: Normal (N), Diabetes (D), Glaucoma (G), Cataract (C), Age-related Macular Degeneration (AMD), Hypertension (H), Myopia (M), and Other diseases/abnormalities (O). These categories encompass many common ophthalmic diseases and present an opportunity to explore the long-tail problem of datasets where some categories have much less data than others.

The significance of developing local datasets like ODIR-5K is profound for constructing medical diagnostic AI that is more suitable for the Chinese population, addressing biases that datasets can bring into AI research. By tailoring datasets to reflect the demographic and clinical profiles of the local population, AI systems can deliver more accurate and reliable diagnoses, ultimately enhancing patient care.

Table 7: Details about ODIR-5K dataset

| Tag | Training Set | Off-Site Test Samples | On-Site Test Samples | Total |
|-----|-----|-----|-----|-----|
| N | 1135 | 161 | 324 | 1620 |
| D | 1131 | 162 | 323 | 1616 |
| G | 207 | 30 | 58 | 307 |
| C | 211 | 32 | 64 | 243 |
| A | 171 | 25 | 47 | 295 |
| H | 94 | 14 | 30 | 138 |
| M | 177 | 23 | 49 | 249 |
| O | 944 | 134 | 268 | 1346 |

## A.2 RFMiD

The RFMiD 2.0 dataset is a multi-label classification dataset for fundus images. It is an updated version of the RFMiD (1.0 version), which was used in the RIADD Challenge at ISBI 2021. Released in 2023, the update primarily includes modifications to the label categories, refining the previously general 'Other' category into specific rare diseases, thereby enhancing the quality of the labels. The dataset contains a total of 3,200 cases, and as of now, images and labels for all cases have been provided. With annotations for 45 different ocular diseases, RFMiD 2.0 holds the distinction of having the most disease categories among publicly available fundus datasets.

Fundus images, due to their similarity to natural images, have been used as a benchmark for the generalizability of methods in the medical field by researchers of natural images. This includes datasets such as CHASE, DRIVE, etc., which have been discussed in our previous articles. As the fundus dataset with the most annotated categories available publicly, RFMiD 2.0 is of value to both general multi-label classification researchers and those working on computer-aided diagnostic research related to vision. Given the large number of annotation categories in the dataset, some categories inevitably have very few samples, making this dataset also relevant for research into multi-label long-tail problems and issues related to limited samples in the medical field.

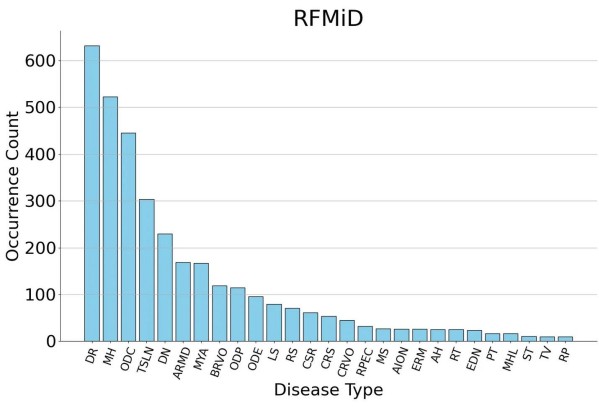

Figure 2: Class count of data points in the RFMiD Dataset

## A.3   JSIEC

The JSIEC dataset consists of a total of 209,494 fundus images, covering 39 categories. This article introduces a subset of the JSIEC dataset, which includes 1,000 fundus images distributed across 39 categories. This dataset collects fundus images from 7 different data sources for the development and validation of deep learning algorithms. The primary datasets for training, validation, and testing come from the Picture Archiving and Communication System (PACS) of the Joint Shantou International Eye Center (JSIEC) in China, China's Lifeline Express Diabetic Retinopathy Screening System (LEDRS), and the Eye Picture Archiving and Communication System (EyePACS) in the USA.

Millions of people worldwide are affected by fundus diseases such as Diabetic Retinopathy (DR), Age-related Macular Degeneration (AMD), Retinal Vein Occlusion (RVO), Retinal Artery Occlusion (RAO), Glaucoma, Retinal Detachment (RD), and fundus tumors. Among these, DR, AMD, and Glaucoma are the most common causes of vision impairment in most populations. Without accurate diagnosis and timely appropriate treatment, these fundus diseases can lead to irreversible blurring of vision, visual distortion, field defects, and even blindness. However, in rural and remote areas, especially in developing countries, there is a lack of ophthalmic services and ophthalmologists, making early detection and timely referral for treatment often inaccessible. Notably, fundus photography provides basic detection of these diseases and is available and affordable in most parts of the world. Non-professionals can handle fundus photographs and send them online to major ophthalmic institutions for follow-up. Artificial intelligence technology can be used to assist in diagnosis.

## B   Images

Table 8: Details about the JSIEC dataset

| Disease | No. of Images | Disease | No. of Images |
|---|---|---|---|
| Normal | 38 | CRVO | 22 |
| Tessellated | 13 | Yellow-white spots-flecks | 29 |
| Large Optic Cup | 50 | Cotton-wool spots | 10 |
| DR1 | 18 | Vessel tortuosity | 14 |
| Possible glaucoma | 13 | Chorioretinal atrophy-coloboma | 15 |
| Optic atrophy | 12 | Preretinal hemorrhage | 10 |
| DR2 | 49 | Fibrosis | 10 |
| DR3 | 39 | Laser Spots | 20 |
| Sever hypertensive | 15 | Silicon oil in eye | 19 |
| Disc swelling and elevation | 13 | Blur fundus without PDR | 111 |
| Dragged Disc | 10 | Blur fundus with suspected PDR | 45 |
| Congenital disc abnormality | 10 | RAO | 16 |
| Retinitis pigmentosa | 22 | Rhegmatogenous RD | 57 |
| Bietti Crystalline Dystrophy | 8 | CSCR | 14 |
| Peripheral Retinal Degeneration and Break | 14 | VKH Disease | 14 |
| Myelinated Nerve Fiber | 11 | Maculopathy | 74 |
| Vitreous Particles | 14 | ERM | 26 |
| Fundus Neoplasm | 8 | MH | 23 |
| BRVO | 44 | Pathological Myopia | 54 |
| Massive Hard Exudates | 13 | | |

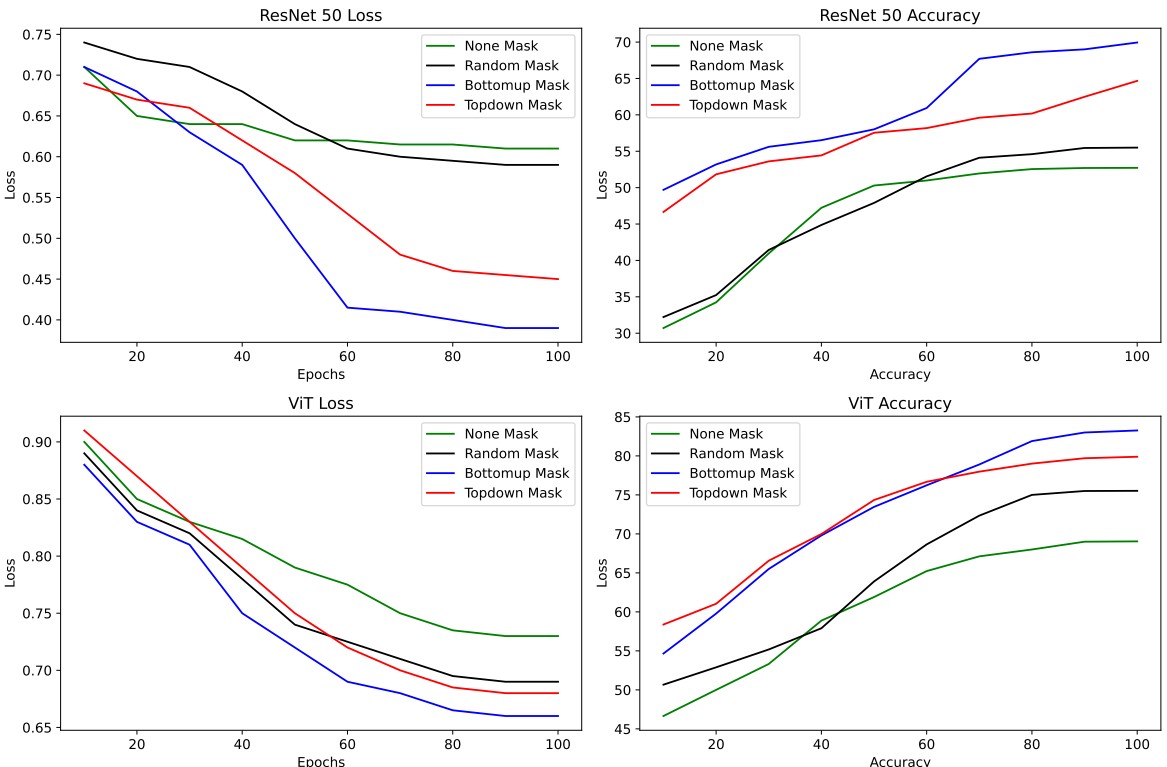

Figure 3: These plots show the loss curves and accuracy curves for the different models used. The top row has the metrics for ResNet18 model, and the bottom row has the metrics for the Vision Transformer model. We also plot metrics for each of the masks evaluated: `none`, `random`, `topdown` and `bottomup`.

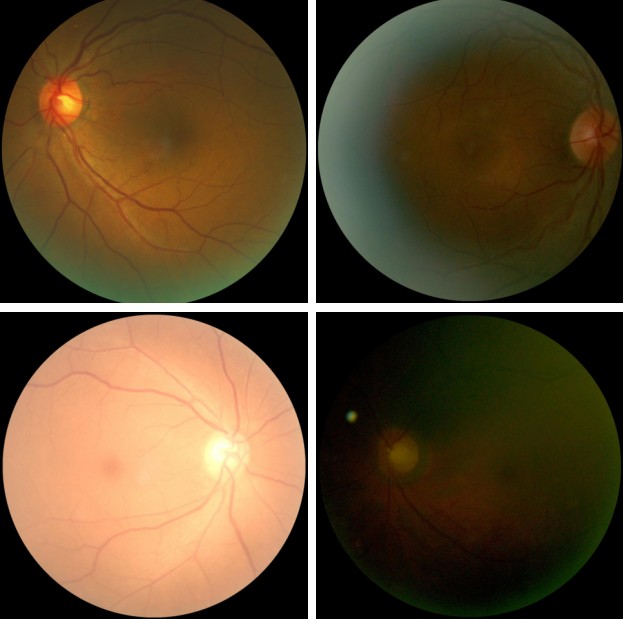

Figure 4: Fundus images from datasets with the minimum and maximum entropy. The top row consists of diabetic and normal fundus images, respectively, which have the minimum entropy. The bottom row consists of diabetic and normal fundus images, respectively, which has maximum entropy. We note that the model has highest confidence in its predictions when the image is clear, and the least confidence when the image is under or over-exposed.

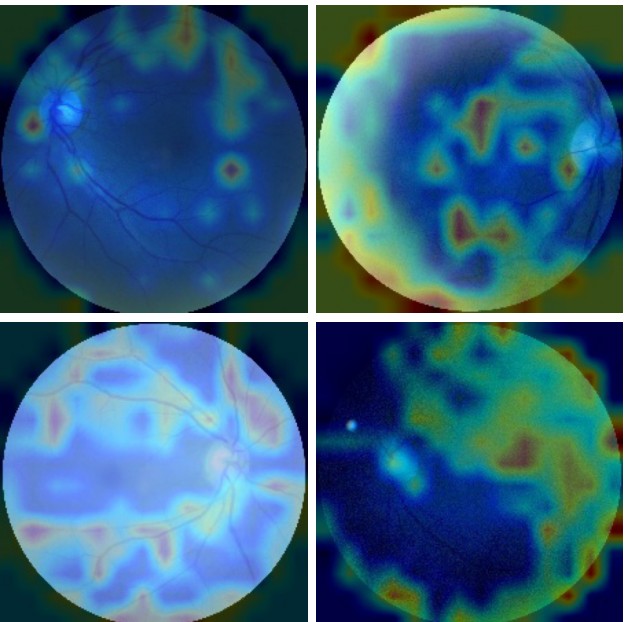

Figure 5: GradCAM analysis of the attention maps of the Vision Transformer. The top row consists of fundus images of diabetic and normal patients with minimum entropy. The bottom row consists of fundus images of diabetic and normal patients with maximum entropy. On top of these images, we apply the attention map computed using GradCAM to understand which parts are considered important by the model.

