# OpenReview forum: "Enhance Eye Disease Detection using Learnable Probabilistic Discrete Latents in Machine Learning Architectures"
_TMLR — Rejected by TMLR_

### Review · Reviewer_hFoY · 2024-12-11

**Summary Of Contributions:**

The authors investigate the utility of generative flow nets (GFlowNets) in improving the diagnostic accuracy and uncertainty estimation of resnets and vision transformers trained to identify diabetic retinopathy in fundus images.

They test 4 different dropout mask generation strategies - no dropout, standard dropout, bottom-up, and top-down - and find the bottom up strategy to be most effective, leading to significant improvements in classification accuracy over standard dropout, for both the resnet-18 as well as ViT models. This improvement extends to the experimental setting where gaussian noise is applied to the images, as well as a held out dataset with noise added to it.

Finally, they show qualitative examples of images with high and low entropy (as computed from their models) to highlight the uncertainty in predictions when the fundus image is under- or over-exposed. Qualitative examples of gradCAM attention maps are also provided to explain model outputs.

**Audience:**

No

**Claims And Evidence:**

No

**Requested Changes:**

1. Please provide code implementations for verifying the experimental results presented in the paper.
2. It is important to report external baselines on the ODIR dataset, ideally the current state-of-the-art method.
3. Claims about better accuracy would need to be demonstrated across multiple disease conditions, and ideally across multiple datasets and vision backbones, to unequivocally show that the improvement in performance due to the GFLowOut layers is indeed generalizable.
4. Claims about improved uncertainty estimation would need to be substantiated with experiments on OOD detection, again, ideally across multiple other fundus imaging datasets and disease conditions.
5. A detailed mathematical overview of GFLowOut and how it has been adapted for this work would be appreciated by readers who are not familiar with this specific line of research.
6. Detailed descriptions of the models used and optimization methods employed for training the newly inserted layers are necessary to fully understand and replicate this work.
7. Please include threshold independent measures of model performance such as AUROC. The average precision would be even better since this is an imbalanced classification task.

Each of these proposed changes is critical for recommending acceptance.

**Strengths And Weaknesses:**

**Strengths**

GFlowOut has been used for natural image classification tasks to improve prediction accuracy and better quantify uncertainty. This paper attempts to test their efficacy for medical imaging, specifically ocular imaging.

**Weaknesses**
The main claims of this paper are not well substantiated by the experimental results reported. These results are not contextualized by any external baselines and the improvements reported by using GFlowOut (bottom-up and top-down strategies) are questionable. An overall lack of detail in the description of the experimental methods makes it challenging to independently validate these results. Even if they are taken as is, there is insufficient evidence that the insights from this paper are generalizable outside this narrow application domain.

1. *Claims*:
    - One of the key contributions claimed is that GFlowOut improves accuracy across a diverse set of ocular conditions, however experiments are only performed on one condition (diabetic retinopathy) despite the dataset having 7 unique conditions.
    - The second claim is that GFlowOut improves uncertainty estimation. This claim could have been substantiated by running experiments on an OOD detection task, similar to the original GFLowNet paper, but no such evidence is presented here. Note that showing better generalization on OOD (as shown here) is evidence of robustness, not evidence of better uncertainty estimation.

2. *Methodology*:
	- There are a number of other studies referencing the ODIR dataset used in these experiments, but there is no discussion of any competing methods to establish a baseline. Given this absence, merely showing improvements on resnet-18 and a ViT of unspecified size is not particularly convincing, since the baselines implemented by the authors might not be very strong.
	- There is strong reason to question the validity of their baselines. Given that there are 2873 normal images and 1608 diabetes images, even a naive model that always predicts the majority class would score an accuracy of 64%, over 10% more than some of the reported resnet-18 baselines.
	- Even the original GFLowOut paper reports only modest improvements over existing dropout methods, so the stark improvements reported by the authors, while not impossible, are rather unexpected.
	- The entropy values mentioned in table 6 provide little insight and the qualitative examples of in figure 4 and 5 do not really move the needle in terms of evidence needed to justify the claims presented, not do the loss curves in figure 3.

3. *Lack of details*:
	- Given that there is no code implementation provided to independently run these experiments, it is hard to verify these results.
	- There is no discussion regarding the specific mathematical details of GFLowOut or how it works, or how the "Trajectory Balance objective" or "reward function" are related to the binary image classification task explored in this paper.
	- There is no explanation of how the GFLowOut layers inserted into the vision backbones are trained. Section 3.1 merely says the backbone is frozen and the final classification head is trained. Without any clear indication of the training objective for these inserted layers, it's impossible to figure out how they were optimized.
	- Figure 1 is rather ambiguous with no details on the GFlowOut layer or what the depicted distribution is supposed to represent. Nor is there any depiction of the resnet-18 also used in this paper.

---

### Review · Reviewer_VXuG · 2024-12-15

**Summary Of Contributions:**

The submission applies GFlowOut to ResNet-18 and Vision Transformer pre-trained on imagenet. The networks are fine-tuned on the ODIR dataset for binary classification normal/diabetes. Different dropout conditioning is compared on the original images, as well as on images with added noise. The trained classifiers are also evaluated on an out-of-distribution dataset (JSIEC). The authors claim to achieve better performance than with traditional dropout.

**Audience:**

No

**Broader Impact Concerns:**

No concerns.

**Claims And Evidence:**

No

**Requested Changes:**

Critical changes:

1. If the authors want to show that "bottomup" and/or "topdown" improve accuracy, then they need to show this with convincing experiments. Critically, they need to compare to strong baseline numbers to make sure that the baselines have been properly tuned. Otherwise it is not possible to say whether a change in reported results is due to better methodology, or mistakes in the comparison numbers. The baselines should either be from trustworthy published results, or the authors need to show strong results in their own setup with convincing arguments that all runs are properly tuned for best performance.
1. If the authors want to show that the "model uncertainty estimation is improved", they need to do an analysis of the model uncertainty. Section 4.2 does not provide any details to the method used for "entropy calculation", it does not specify how "uncertainty estimation" is different in the various dropout flavors presented in the paper, and it's not clear what could be learnt from a single GradCAM visualization, and what could be learnt from the min/max/avg entropy comparison over the OTIR vs. JSIEC datasets.
1. The data used needs to be documented better. The Kaggle link for the OTIR dataset, and the zenodo link to the JSIEC dataset are insufficient. The authors should provide references to publications that properly characterize the data used, including provenance of the data, and licensing terms.
1. If the authors want to make claims about practical usefulness of their method, they need to make it clearer why any of the presented findings have practical significance. For example, the datasets and performance of the model need to be more realistic. Similarly, they should focus on an uncertainty setup that has practical significance (e.g. splitting the train/test set by location, instead of using random splits or adding procedural noise).

Additional changes:

1. The citations are malformatted: e.g. "Wong et al. (2014)" should probably be "(Wong et al., 2014)" – this comment applies to whole document
1. Introduction: "primarily focuse" should be "primarily focus"
1. Introduction: "There are multiple works shown" should be "There are multiple works showing"
1. Introduction: "artificial intelligence(AI)" should be "artificial intelligence (AI)"
1. Introduction: not clear what "their performance can be limited by using uniform image sizes" means
1. Figure 1: it seems that there is a single transformer layer, and its output goes into a network called GFlowOut – but Section 3.1 says that every dropout layer is replaced by a dropout layer that is modeled by a GFlowNet – this is also how it is explained in [(Liu et al., 2023)](https://arxiv.org/abs/2210.12928). Similarly, it's unclear what "MLP Heads" refer to.
1. Eye Disease Dataset: why crop to 224px and then resize to 256px? maybe they were first resized to 256px and then center-cropped to 224px? also, not clear what the sentence "finally, bi-linear interpolation was applied" means...
1. Section 4.1: "The Vision Transformer, being both a larger and transformer based model as compared to the ResNet-50 model" – otherwise, the submission always mentions ResNet-18, not ResNet-50
1. Figure 4: right images ("normal") both seem to be very low quality; I would expect the model to have uncertain predictions for both of these
1. Figure 5: it's unclear what should be illustrated with these four images

**Strengths And Weaknesses:**

Strengths

1. The paper focuses on a topic (eye disease detection from retinoscopy images) with global significance.
1. Attempt to better understand uncertainty in clinical applications.

Weaknesses

1. I don't think that the presented results allow us to draw the conclusions presented in the paper. The reported numbers are very low (e.g. accuracy 52.72% for ResNet, 69.04% for Vision Transformer). The four comparison dropout methods (none, random, bottomup, topdown) could simply be tuned insufficiently (e.g. too little random dropout), which might explain the presented differences, rather than one method performing better than the others, as the authors claim.
1. It seems that the dataset that is used is very small (2873 normal images, 1608 diabetes images – it's unclear what grade of diabetic retinopathy is referred to). For example, already in [(Gulshan, 2016)](https://jamanetwork.com/journals/jama/fullarticle/2588763), a classifier was trained on more than 128k images for a similar application. This might explain at least in part the low performance reported, and this severely limits the utility of the study. Furthermore, the datasets (OTIR, JSIEC) have practically no documentation (on the Kaggle/zenodo webpages).
1. It's unclear what is the significance of Section 4.2 – the model is evaluated on JSIEC, but the result of the evaluation (e.g. accuracy) is not mentioned. Table 6 shows that there are different entropies calculated for different datasets, but it's unclear what that means. Figures 4 and 5 show a single example of a normal/diabetic fundus image with minimum/maximum entropy, but it's unclear what could be learnt from these examples.

---

### Review · Reviewer_d1ss · 2024-12-26

**Summary Of Contributions:**

The paper performs ocular disease classification using ResNet / Vision Transformer (ViT) neural networks and  uncertainty quantification (UQ) with GFlowOut. The paper finds that ViT performs better than ResNet and that GFlowOut outperforms traditional dropout.

**Audience:**

Yes

**Claims And Evidence:**

Yes

**Requested Changes:**

1. Why did the authors choose the forms of noise they did? Is there real-world justification for Gaussian, Speckle, and Salt?
2. The entropy analysis would be improved with some sort of quantitative analysis, e.g. measuring how correlated entropy is to accuracy empirically.

Minor comments:
1. Make sure to double check that relevant citations are in parentheses.
2. Figs 4 and 5 take up a lot of white space. Consider including additional images or re-arranging to take up less vertical space.

**Strengths And Weaknesses:**

1. The paper performs a convincing empirical evaluation of GFlowOut on the important application of ocular disease classification from fundus images, showcasing the benefit of this recently proposed method for UQ.
2. The paper tests different masking strategies which can effectively serve as ablations for the benefit of leveraging additional information in the conditioning set for the variational distribution.

---

### Author Response · Authors · 2025-01-07
**Thank you for the feedback.**

Dear Reviewers,

Thank you for taking the time to review our manuscript. We greatly appreciate the effort and thoughtful feedback you have provided. Your comments and suggestions are invaluable in helping us improve the quality and clarity of our work.

We are currently addressing your comments and will update the manuscript accordingly. You can expect the revised version and response to be submitted within the next 2–3 days.

Thank you once again for your constructive input and for contributing to the refinement of our research.

---

### Author Response · Authors · 2025-01-11
**Response to Reviewers - Summary of Changes**

Dear Reviewers,

Thank you for taking the time to review our manuscript “Enhance Eye Disease Detection using Learnable Probabilistic Discrete Latents in Machine Learning Architectures.” We sincerely appreciate your thoughtful feedback and constructive suggestions, which have been invaluable in improving the clarity and quality of our paper. We have carefully considered each of your comments and incorporated changes to address them. Below, you will find a summary of the changes, a detailed point-by-point response to your feedback, and the new manuscript included for your kind review.

---

## Summary of Changes

We have made significant revisions to the manuscript in response to reviewer feedback. Below is a summary of the changes implemented:

### 1. Experiments on New Dataset (RFMiD)
- Added experiments on the Retinal Fundus Multi-Disease (RFMiD) dataset to assess the generalizability of the proposed method across multiple ocular diseases.
- Results from RFMiD are included in the revised tables, demonstrating the robustness of GFlowOut in handling diverse datasets.

### 2. Updated Model Architecture Visualization (Figure 1)
- Revised Figure 1 to provide a clearer and more detailed representation of the model architecture.
- Enhanced labeling and annotations to improve readability and illustrate the integration of GFlowOut into ResNet18 and Vision Transformer backbones.

### 3. Added Algorithm Section
- Introduced a new Algorithm section with pseudocode (Algorithm 1) detailing the step-by-step process of model implementation, from data preprocessing to disease classification.
- Added the mathematical details of GFlowOut and how it works, including the relationship between the "Trajectory Balance objective" or "reward function" and the image classification task explored in the paper.

### 4. Additional Metrics
- Expanded tables and figures for metrics such as precision, recall, F1-score, accuracy, and AUROC.
- Provided weighted averages for metrics across datasets.

### 5. Uncertainty Measurements Using Entropy
- Added entropy mean and variance calculations to quantify prediction uncertainty.
- Results are included in updated tables and discussed in the context of robustness to out-of-distribution (OOD) data.

### 6. Added External Baselines/Benchmarks
- Incorporated external results as baselines and references from Li et al. (2021) and Gulshan et al. (2016) to provide better context to our work:
  - **Baseline from Li et al. (2021):** Incorporated results from *"A benchmark of ocular disease intelligent recognition: One shot for multi-disease detection"* by Li et al. (2021). Their approach benchmarks the ODIR dataset for multi-disease classification and provides a meaningful reference for evaluating our method.
  - **Reference to Gulshan et al. (2016):** The introduction section has been updated to include:
    > "Early diagnosis and effective management are crucial in preventing these diseases from progressing to more severe stages. Traditional diagnostic methods, which typically involve manual examination of retinal images, are often time-consuming and subject to variability among practitioners. Since 2016, Google applied Deep Learning to analyze retinal images for the detection of diabetic retinopathy (Gulshan et al., 2016). The utilization of Deep Learning has shown immense promise in automating the analysis of medical images, providing more consistent and scalable solutions for disease diagnosis. However, issues around the reliability of these models and their capacity to estimate uncertainty continue to present challenges in clinical decision-making (Li et al., 2022)."

### 7. Justified Noise and Entropy Analysis
- Added explanations for using Gaussian, Salt-and-Pepper, and Speckle noise in experiments.
- Justified the role of entropy analysis in evaluating model robustness and highlighted its utility in clinical scenarios.

### 8. Formatting and Typographical Improvements
- Fixed minor typographic errors and reference inconsistencies in formatting.
- Improved figure and table captions for better comprehension.
- Reformatted mathematical equations and algorithm pseudocode for readability.

---

### Author Response · Authors · 2025-01-11
**Response to Reviewer VXuG**

**Response to Reviewer VXuG**

**1. The data used needs to be documented better. The Kaggle link for the OTIR dataset, and the Zenodo link to the JSIEC dataset are insufficient. The authors should provide references to publications that properly characterize the data used, including the provenance of the data and licensing terms.**

**Response:**
We thank the reviewer for pointing out the need for better documentation of the datasets. To address this concern, we have added detailed descriptions of the OTIR and JSIEC datasets, including their provenance, licensing terms, and references to relevant publications, in the appendix section. This additional information provides a more comprehensive overview of the data used in the study.

---


**2. If the authors want to make claims about the practical usefulness of their method, they need to make it clearer why any of the presented findings have practical significance. For example, the datasets and performance of the model need to be more realistic. Similarly, they should focus on an uncertainty setup that has practical significance (e.g., splitting the train/test set by location instead of using random splits or adding procedural noise).**

**Response:**
We appreciate the reviewer’s feedback regarding the practical significance of our method. To emulate clinical conditions, we have intentionally used multiple datasets encompassing a variety of medical conditions. Additionally, running experiments with added noise helps simulate imperfections that may arise during the imaging process, making our evaluation more reflective of real-world scenarios. However, we acknowledge the limitation of the dataset, which does not include metadata such as location, age, or gender. These elements could indeed have provided a more meaningful basis for train/test splits compared to the random splitting approach used. We will continue to explore opportunities to incorporate such richer datasets in future work.

---


**3. Figure 1: It seems that there is a single transformer layer, and its output goes into a network called GFlowOut – but Section 3.1 says that every dropout layer is replaced by a dropout layer that is modeled by a GFlowNet – this is also how it is explained in (Liu et al., 2023). Similarly, it's unclear what "MLP Heads" refer to.**

**Response:**
We appreciate the reviewer’s feedback and have revised Figure 1 to more accurately reflect the architecture described in Section 3.1. The updated figure and its corresponding caption now provide a clearer representation of the integration of GFlowOut within the Vision Transformer (ViT) architecture for eye disease detection.

---


**4. Eye Disease Dataset: Why crop to 224px and then resize to 256px? Maybe they were first resized to 256px and then center-cropped to 224px? Also, it is unclear what the sentence "finally, bi-linear interpolation was applied" means.**

**Response:**
We appreciate the reviewer’s observation and apologize for the confusion in the previous version. This has now been corrected. We clarify that we are using the inbuilt preprocessing pipeline provided by the `torchvision` module for image preprocessing. Specifically:
- The images are first resized to 256px and then center-cropped to 224px.
- The mention of "bi-linear interpolation" refers to the resampling method used during the resizing step, as implemented in the `torchvision.transforms.Resize` function.

We have updated the manuscript to reflect this clarification.

---


**5. Section 4.1: "The Vision Transformer, being both a larger and transformer-based model as compared to the ResNet-50 model" – otherwise, the submission always mentions ResNet-18, not ResNet-50.**

**Response:**
We thank the reviewer for pointing out this inconsistency. The reference to "ResNet-50" in Section 4.1 was incorrect and has been updated to "ResNet-18" to ensure consistency throughout the submission.

The corrected sentence now reads:
> *"The Vision Transformer, being both a larger and transformer-based model as compared to the ResNet-18 model..."*

---


**6. Figure 4: The right images ("normal") both seem to be very low quality; I would expect the model to have uncertain predictions for both of these.**

**Response:**
We appreciate the reviewer’s observation regarding the quality of the images in Figure 4. The images were randomly selected from the dataset, and the perceived lower quality may be due to scaling issues that occurred during the PDF generation process. To address this, we will ensure that the images are properly scaled and of higher resolution in the revised submission. Thank you for bringing this to our attention.

---

### Author Response · Authors · 2025-01-11
**Response to Reviewer d1ss**

**Response to Reviewer d1ss**

**1.Why did the authors choose the forms of noise they did? Is there real-world justification for Gaussian, Speckle, and Salt?**

**Response:**
We thank the reviewer for this insightful question. We chose Gaussian, Speckle, and Salt noise because these forms of noise closely simulate real-world challenges encountered in medical imaging, particularly in clinical settings. Each type of noise represents specific scenarios:
- Gaussian noise models random variations in image intensity, often arising from electronic sensor noise.
- Speckle noise  is common in medical imaging modalities like ultrasound, where coherent processing leads to granular interference patterns.
- Salt noise (impulse noise) represents sudden, extreme pixel intensity variations, which can occur due to data transmission errors or sensor faults.
These noise types ensure the robustness of our model under various real-world conditions. We have expanded on this justification in the revised manuscript (see Section 3.3). Please let us know if further clarification is required.

---
**2. The entropy analysis would be improved with some sort of quantitative analysis, e.g., measuring how correlated entropy is to accuracy empirically.**

**Response:**
We appreciate the reviewer’s suggestion regarding quantitative analysis of entropy and its correlation with accuracy. In our experiments, entropy is used as a measure of uncertainty. While we have not directly correlated entropy with accuracy, we compare the entropy measures of our models against baseline models to demonstrate improved uncertainty estimation. These comparisons highlight the superior performance of our models in capturing uncertainty.

We acknowledge the value of empirically analyzing the correlation between entropy and accuracy and will consider incorporating such analyses in future work to further strengthen the evaluation.

---
**3. Make sure to double-check that relevant citations are in parentheses. Figs. 4 and 5 take up a lot of white space. Consider including additional images or re-arranging to take up less vertical space.**

**Response:**
We appreciate the reviewer’s suggestions and have taken the following actions to address these points:
1. All citations have been reviewed and corrected to ensure they are properly enclosed in parentheses.
2. Figures 4 and 5 have been resized and rearranged to minimize white space and improve the overall layout. We have also considered including additional images where appropriate to make better use of the space.

---

> ### Comment · Reviewer_d1ss · 2025-01-11
> **Reviewer response**
>
> Thank you for the response. I'd like to emphasize that my suggestion for an improved entropy analysis is because a primary argument in the paper is that GFlowOut offers better uncertainty estimation, and this claim needs stronger evidence. As it currently stands, I do not see how this claim is clearly supported by the paper's analysis--for instance, Table 5 only shows that OOD datasets have higher average entropy under the proposed setting than under the baseline setting, but this need not mean the former's estimate is more accurate, nor even that OOD detection performance is better; actually testing OOD detection directly would be better to make the latter claim, for instance. The qualitative analysis starts to get at this notion of better uncertainty estimation (e.g., with the statement, "Images with the highest entropy values...tend to perform worst), but a quantitative analysis would be much more convincing. My previous suggestion of correlating entropy with accuracy was inspired by calibration metrics such as expected calibration error (ECE), though they are equivalent for the right binning choice.

---

> > ### Author Response · Authors · 2025-01-13
> > **Response to Reviewer d1ss**
> >
> > Response:
> > Dear Reviewer,
> >
> > Thank you for your detailed feedback and suggestions. We appreciate your emphasis on providing stronger evidence for the claim that GFlowOut improves uncertainty estimation. To address this, we conducted additional experiments , including an evaluation using Expected Calibration Error (ECE) as advised and we getting good result. We are in the process of finalizing these updates and will incorporate them into the manuscript by the end of the day. We will notify you once the updated draft is available.
> >
> > Thank you again for your valuable suggestions, which have helped us refine this critical aspect of the paper.

---

> > ### Author Response · Authors · 2025-01-14
> > **Response to Reviewer d1ss**
> >
> > **Response:**
> >
> > Dear Reviewer,
> >
> > We appreciate your insightful suggestions, which have significantly strengthened this aspect of our paper.  We have updated our manuscript to include a detailed analysis using Expected Calibration Error (ECE) as a quantitative metric.
> >
> > 1. Updates to Manuscript:
> >    - In Section 3.4 (Out-of-Distribution Evaluation and Entropy Calculations), we now include a thorough discussion of ECE as a calibration metric to assess uncertainty estimation.
> >    - In Table 6, we present the ECE scores for both In-Distribution (ID)  and Out-of-Distribution (OOD) datasets. The ECE scores are calculated for the baseline and GFlowOut models, using \( M = 10 \) bins for calibration evaluation.
> >
> > 2. Key Findings:
> >    - The results in Table 6 demonstrate that GFlowOut achieves significantly lower ECE scores than the baseline model for both ID and OOD datasets.
> >    - This improvement highlights that GFlowOut produces better-calibrated predictions, ensuring that the model's predicted probabilities align more closely with observed accuracies.

---

### Author Response · Authors · 2025-01-11
**Response to Reviewer hFoY- part 1**

**Response to Reviewer hFoY**


### 1. The main claims of this paper are not well substantiated by the experimental results reported. These results are not contextualized by any external baselines, and the improvements reported by using GFlowOut (bottom-up and top-down strategies) are questionable.

**Response:**
We have included external baselines from established works, specifically:

- Baseline from Li et al. (2021):
  We incorporated results from *“A benchmark of ocular disease intelligent recognition: One shot for multi-disease detection”* by Li et al. (2021). Their approach benchmarks the ODIR dataset for multi-disease classification and provides a meaningful reference for evaluating our method.
  - Where applicable, we replicated their experimental settings for direct comparability.
  - Our results are presented alongside their benchmarks in our revised manuscript, highlighting the improvements achieved using GFlowOut.

- Reference to Gulshan et al. (2016):
  We have also referenced the pioneering work by Gulshan et al. (2016), which introduced a deep learning-based method for diabetic retinopathy detection using retinal images. While their dataset differs from ODIR, their findings provide valuable context for evaluating the significance of our contributions.

---

### 2. An overall lack of detail in the description of the experimental methods makes it challenging to independently validate these results.

**Response:**
To address this comment, we have significantly enhanced the descriptions in the methods and experiments sections. These revisions now include:
- A detailed breakdown of data preprocessing steps, model architectures, and training protocols.
- A step-by-step pseudocode implementation (*Algorithm 1*) to guide independent replication of the experiments.
- Expanded Section 3.2 to include a detailed mathematical formulation of GFlowOut and its integration with our models.
  Below is a summary of the key mathematical details added to the manuscript.

---

### 3. One of the key contributions claimed is that GFlowOut improves accuracy across a diverse set of ocular conditions; however, experiments are only performed on one condition (diabetic retinopathy) despite the dataset having 7 unique conditions.

**Response:**
We have addressed this concern by extending our experiments to include all the classes in the original dataset. Additionally, we incorporated the RFMiD dataset as an additional external dataset. This dataset contains a diverse range of ocular conditions, enabling us to evaluate the generalizability of GFlowOut across different diseases and imaging modalities.

---

---

### Author Response · Authors · 2025-01-11
**Response to Reviewer hFoY- part 2**

### 4. Provide code implementations for verifying the experimental results presented in the paper.

**Response:**
We did not include the GitHub link in our submission because the paper submission instructions stated that it should be anonymous, without author names. We have updated the abstract to include the GitHub link. The updated text is as follows:
> The source code for all these experiments can be found at https://github.com/anirudhprabhakaran3/gflowout_on_eye_images

---

### 5. There is strong reason to question the validity of their baselines. Given that there are 2873 normal images and 1608 diabetes images, even a naive model that always predicts the majority class would score an accuracy of 64%, over 10% more than some of the reported ResNet-18 baselines.

**Response:**
We acknowledge the limitations of relying solely on accuracy, particularly for imbalanced datasets such as ODIR. To address this concern:
- We have incorporated additional threshold-independent metrics to provide a more comprehensive evaluation of model performance, including  AUROC, F1-Score, Precision, and Recall.
- We re-ran the experiments with the hyperparameters from the baseline papers, ensuring that our models at least match, if not improve, on the external baselines.
---

### 6. A detailed mathematical overview of GFlowOut and how it has been adapted for this work would be appreciated by readers who are not familiar with this specific line of research.

**Response:**
Thank you for your valuable suggestion. To address this comment, we have significantly expanded the mathematical foundations of GFlowOut and its integration into our work. The following updates have been made:

1. Detailed Mathematical Overview:
   - We added a comprehensive explanation of the  flow consistency equation ,  trajectory balance objective , and  posterior distribution over dropout masks in Section X of the revised manuscript.
   - This includes key equations, such as:
     - The flow consistency equation:
       $$ F(s) = \sum_{a \in \text{Actions}(s)} F(s, a) $$
     - The trajectory balance objective:
       $$ \prod_{t=0}^{T-1} P(a_t \mid s_t) = \frac{R(s_T)}{Z} $$
     - Posterior distribution computation using a softmax function:
       $$ P(\text{dropout} \mid X, H) = \text{softmax}(W \cdot [X, H]) $$
   - These equations clarify how GFlowOut generates dropout masks in proportion to a task-specific reward function, improving robustness and uncertainty estimation.

2. Adaptation for This Work:
   - We explained how GFlowOut is integrated into ResNet18 and Vision Transformer architectures by replacing standard dropout layers with learnable probabilistic dropout layers.
   - The description includes details on how dropout probabilities are conditioned on contextual features from preceding layers, enhancing both accuracy and uncertainty estimation.

3. Revamped Model Diagram:
   - The model image (Figure 1) was updated to visually depict the flow of information, the integration of GFlowOut layers, and the overall architecture.
   - A step-by-step pseudocode implementation (*Algorithm 1*) to guide independent replication of the experiments.

These additions aim to make the paper accessible to a broader audience while maintaining technical rigor. The revised text and updated figure are now included in the manuscript.

---

### Decision · Action_Editor_oJUv · 2025-01-27

**Recommendation:** Reject

**Comment:**

There has been an active discussion between the authors and reviewers during the rebuttal phase. However, the reviewers have not been convinced by the authors arguments and their revised manuscript.

**Audience:**

The reviewers agree that the paper takes an existing method and attempts to apply it to fundus image datasets. Even if the experiments had been conducted more rigorously, they are doubtful whether the results would have been of interest to the TMLR audience. They find that the paper focuses on applying existing methodology (i.e., GFlowOut) to a specific application (i.e., medical imaging) with minimal methodological or application-specific improvements. The uncertainty evaluations (via entropy calculations) are also rather superficial for the claim that the proposed setting yields better uncertainty estimates. The reviewers further find that in its current form, the paper might be better served in a machine learning for healthcare venue, where the introduction of GFlowOut to the community could be impactful. Otherwise, they believe a more comprehensive analysis and evaluation, as well as additional innovations, are necessary to recommend publication to TMLR.

**Claims And Evidence:**

After thorough discussion, all reviewers agree that the claims of the paper are not supported by sufficient evidence. The reviewers find that the paper does not give any evidence that "learnable probabilistic latents significantly improves accuracy", as opposed to comparing the method and other methods ("none", "random") in an insufficiently tuned setting. They also don't think the few examples of Grad-CAM show that the model "accurately focuses on critical image regions". Moreover, a main argument of the paper is that the GFlowOut can offer better uncertainty estimation, but the entropy evaluations only minimally support this claim (e.g., they only show that the proposed setting assigns higher entropy to OOD images on average than the baseline setting). The revised paper merely reports average entropy values for an OOD dataset. There is no serious attempt to validate this claim on an OOD detection task unlike the original GFlowOut paper. Lastly, the results for F1 score reported in [1] for a variety of backbones (including resnet-18) are much higher than the ones reported by the authors in their paper. Other works such as [2] similarly report much higher numbers for resnet-18 and other backbones. In addition, the revision of the manuscript changed all the numbers in Table 2 (e.g. the "none" ResNet18 Accuracy was 52.72 in the original manuscript, but then it suddenly changed to 61.08 in the updated version. These discrepancies do not inspire confidence in the validity of this paper's claims.


[1] A Benchmark of Ocular Disease Intelligent Recognition: One Shot for Multi-disease Detection

[2] BFENet: A two-stream interaction CNN method for multi-label ophthalmic diseases classification with bilateral fundus images

**Resubmission Of Major Revision:**

The authors may consider submitting a major revision at a later time.